# Replication of Mesoscale Pore One-dimensional Nanostructures: Surface-induced Phase Separation of Polystyrene/Poly(vinyl alcohol) (PS/PVA) Blends

**DOI:** 10.3390/polym11061039

**Published:** 2019-06-12

**Authors:** Paritat Muanchan, Takashi Kurose, Hiroshi Ito

**Affiliations:** 1Graduate School of Organic Materials Science, Yamagata University, 4-3-16 Jonan, Yonezawa, Yamagata 992-8510, Japan; paritat.pm@gmail.com; 2Research Center for GREEN Materials and Advanced Processing (GMAP), Yamagata University, 4-3-16 Jonan, Yonezawa, Yamagata 992-8510, Japan; takashi.kurose@yz.yamagata-u.ac.jp

**Keywords:** Porous Nanostructures, Phase Separation, Thermal Nanoimprint

## Abstract

Mesoscale pore one–dimensional (1D) nanostructures, or vertically aligned porous nanostructures (VAPNs), have attracted attention with their excellent hydrophobic properties, ultra−high surface area, and high friction coefficient, compared to conventional vertically aligned nanostructures (VANs). In this study, we investigate the replication of VAPNs produced by the thermal nanoimprint process using anodic aluminum oxide (AAO^2^) templates (100 nm diameter). Polystyrene/poly(vinyl alcohol) (PS^1^/PVA) blends, prepared by the advanced melt–mixing process with an ultra–high shear rate, are used to investigate the formation of porosity at the nanometer scale. The results reveal that domain size and mass ratios of PVA precursors in the PS matrix play a dominant role in the interfacial interaction behavior between PS^1^–PVA–AAO^2^, on the obtained morphologies of the imprinted nanostructures. With a PVA nanodomain precursor (PS^1^/PVA 90/10 wt%), the integration of PVA nanodroplets on the AAO^2^ wall due to the hydrogen bonding that induces the phase separation between PS^1^–PVA results in the formation of VAPNs after removal of the PVA segment. However, in the case of PVA microdomain precursors (PS^1^/PVA 70/30 wt%), the structure transformation behavior of PS^1^ is induced by the Rayleigh instability between PVA encapsulated around the PS^1^ surfaces, resulting in the PS^1^ nanocolumns transforming into nanopeapods composed of nanorods and nanospheres.

## 1. Introduction

One-dimensional nanostructures inspired by the nature of gecko feet have attracted considerable attention because of their versatile applications such as self-cleaning surfaces, dry adhesives, and for novel applications of thermally and electrically conductive materials [1,2,3,4,5]. Moreover, nanomaterials have recently appeared based on structure-mediated surface functionalities produced by polymer nanoengineering technology [6]. For example, the hydrophobic surface of the cicada wing is covered with natural vertically aligned nanostructures (VANs) which can penetrate and consequently kill *Pseudomonas aeruginosa* within several minutes of adhesion [7,8]. To enhance VANs bacteria-killing performance, the characteristics and surface properties of VANs need improvement, including their (1) hydrophobic properties, (2) surface areas, and (3) bacteria adherence (ex. surface friction and surface modulus). For this reason, vertically aligned porous nanostructures (VAPNs) have attracted significant attention, with outstanding hydrophobic higher friction properties and ultra-high surface area, because of the nanoscale porosity and surface roughness. Particularly, VAPNs have been used as switchable adhesive surfaces using humidity control [9,10]. Moreover, VAPNs with high value-added have been effectively applied in the fields of energy storage and nanosensing materials [5], and filters [9]. For these reasons, VAPNs are regarded as a novel class of frontier nanomaterials in both industry and academia.

A versatility technique for fabricating VAPNs is the templating method using nanoporous anodic aluminum oxide (AAO) assisted by solvent-wetting and thermal melt-wetting methods. The control of surface-induced phase separation behaviors in the AAO pores is crucial for these techniques. A great deal of research has attempted to improve these methods to be faster, cheaper, more precise, less toxic, and more scalable, which are in conflict with the fundamentals of the solvent-wetting and thermal melt-wetting methods [11,12,13,14]. With the successful development of polymer engineering technologies, the direct patterning method of thermal nanoimprint is an alternative method to replicate the VAPNs because of its high replication capability with rapid speed, zero−solvent use, high precision and reproducibility. For example, one-dimensional polymer nanofiber arrays with various lengths, diameters, aspect ratios, and patterns, were successfully produced by nanoimprint machine using AAO templates. [15]. However, the direct fabrication of VAPNs using the nanoimprint method with AAO templates has not been achieved yet. Therefore, it is a great challenge to develop the thermal nanoimprint process by controlling the phase separation of polymers in the AAO templates.

Study on the various polymer material behaviors at the nanoscale of AAO templates based on the solvent-wetting and thermal melt-wetting methods have been successfully clarified. By using these wetting methods, many publications have reported that polymer behaviors in AAO templates, consist of the flow ability and its gradients, phase separation behaviors, transformation behaviors, and so on. Atsushi Takahara et al [16,17,18] studied flowing gradients by using various polymer blends in AAO templates. With the larger pore size of AAO, it promoted phase separation of polymer blends. In the case of polymer blends with the same value of surface energy, polymer flow gradients were strongly dependent on the glass transition temperature (*T_g_*) and polymer melt viscosity which are related to the polymer chain mobility. At a constant melt-wetting temperature, the obtained nanostructures showed the higher mass ratios of the lower *T_g_* segment due to the thermal-induced polymer chain mobility in the AAO templates. Jiun Tai Chen et al [19] studied the coarsening mechanism of poly(methyl methacrylate) (PMMA) and tetrahydrofuran (THF) confined in AAO templates. In that case, the phase separation of PMMA and THF was induced by the difference in hydrogen bond affinity between PMMA-AAO and THF-AAO. With the stronger hydrogen bond of THF-AAO, the formation of THF droplets on the AAO pore wall was promoted by surface-induced phase separation, resulting in porous PMMA nanostructures after the removal of THF. The formation of porous nanotubes of polystyrene (PS) and fluorescent pyrene-ended PMMA (Py-PMMA) using the solvent-wetting method was also investigated by Jiun Tai Chen et al [20]. Moreover, Thomas P. Russell et al [21] have reported on the Rayleigh instability of PMMA/PS blends confined in AAOs using the solvent-wetting method assisted by thermal annealing. That method can produce PMMA/PS bilayer nanotubes. The thermally-induced Rayleigh instability stimulated the transformation of nanostructures with polymer nanotubes into nanorods and nanospheres. Various transformation and phase separation behaviors of polymer nanomaterials in AAO templates and other atmospheres using chemical solvent and thermal annealing methods have been investigated [22,23,24,25,26,27,28,29,30,31,32,33]. Furthermore, similar polymer phase separations and transformation behaviors at the microscale have also been observed [34,35,36,37,38]. Because of the success of the AAO templating method, many publications have explored various AAO applications such as thermal and physical properties of polymer nanostructures under nanoconfinement conditions, fabrication of higher−order nanostructures, polymerization and crystallization in AAO templates, and the double infiltration nanoencapsulation process [39,40,41,42,43,44,45,46,47,48,49].

In this study, we mainly focus on the formation of VAPNs fabricated using thermal nanoimprint with AAO templates. We investigate the phase separation and the transformation behaviors of polymer blends confined in AAOs. The material models used are polystyrene/poly(vinyl alcohol) (PS/PVA) blends. PS/PVA blends are prepared by the advanced melt-mixing process, using a high-shear machine which effectively produces the ultra-fine dispersion of PVA segments. Several surface properties including surface morphology, hydrophobicity, frictional properties, and the surface mechanical properties of the fabricated VAPNs have been experimentally evaluated. This novel technique has the advantages of being a greener (non-chemical use), rapid process, of high−precision, high−profit, and great reproducibility.

## 2. Materials and Methods 

### 2.1. Materials

Commercial grade PS of two types was used: PS^1^ (GPPS 679; PS Japan Corp., Tokyo, Japan, *T_g_* = 87 °C, melt flow rate (MFR) = 18.0 g/10 min at 200 °C) and PS^2^ (G210; Toyo Styrene Co., Ltd., Tokyo, Japan, *T_g_* = 100 °C, MFR = 10.1 g/10 min at 200 °C). PVA (CP-1210; Kuraray Co., Ltd., Tokyo, Japan, *T_g_* = 26 °C, melting temperature (*T_m_*) = 161 °C, MFR = 4.4 g/10 min at 200 °C) was selectively used.

### 2.2. Preparation of AAOs

High-purity aluminum (Al) sheets (99.99% pure) 5.0 cm × 5.0 cm of 0.5 mm thickness were degreased in acetone, followed by electropolishing in a mixed solution of HClO_4_/H_2_O/EtOH (10.0/7.0/83.0 wt%) at a constant temperature of 5 °C, with 20 V applied for 10 min. The polished aluminum sheets were anodized using a two-step anodization process: The first anodization was done with the conditions shown in Table 1, then the porous alumina was dissolved in a solution containing 6.0 wt% of H_3_PO_4_ and 1.8 wt% of H_2_CrO_4_ at 60 °C for 3 h. The second anodization was applied again with the anodizing conditions shown in Table 1. The remaining aluminum film was removed by dissolving in copper chloride solution. Pore expansion was done by immersion in 8.5 wt% of the H_3_PO_4_ solution at 40 °C. For this study, AAO templates with different two pores size were prepared: AAO^1^ and AAO^2^.

### 2.3. Preparation of Polymer Blends

The PS^1^/PVA pellets with mass ratios of 90/10 and 70/30 wt% were introduced into a high-shear machine (NHSS2-28; Niigata Machine Techno Co., Ltd., Niigata, Japan) for the blending process with a rotation speed of 500 rpm at 180 °C for 10 s. The obtained PS^1^/PVA blends were pelletized with a crushing machine.

### 2.4. Preparation of Polymer Films

The polymer pellets (neat PS^1^, neat PS^2^, and PS^1^/PVA blends) were melted in a template with a thickness of 500 µm at 200 °C for 5 min, followed by pressing 10 MPa in a hot press machine (IMC-11FA; Imoto Machinery Co. Ltd., Tokyo, Japan). The molten film was cooled in a cold press machine (IMC-181B; Imoto Machinery Co., Ltd., Tokyo, Japan) at 5 MPa for 3 min. 

### 2.5. Fabrication of Nanostructures by Thermal Nanoimprint 

In this experiment, one–dimensional nanostructures were fabricated with a thermal nanoimprint machine (Izumi Tech., Miyagi, Japan) using AAO templates. The experimental procedure is shown in Figure 1. The polymer films were placed onto the AAO template and inserted into the thermal nanoimprint machine. The imprinting conditions are presented in Table 2. After imprinting, the AAO was removed by dissolution in 4.0 M of NaOH solution for 24 h, followed by water washing at 60 °C and then the drying process.

### 2.6. Characterization 

Surface morphologies of AAO templates and polymer nanostructures were observed using scanning electron microscopy (SEM, JSM-6510; Jeol Ltd., Tokyo, Japan and TM-1000; Hitachi High-Technologies Corp., Tokyo, Japan) and field emission scanning electron microscopy (FE-SEM, SU-8000; Hitachi High-Technologies Corp., Tokyo, Japan). The surface chemical property of the polymer blends films were analyzed by attenuated total reflection (ATR, AIM-8800 Infrared Microscope; Shimadzu Corp., Kyoto, Japan). The wavenumber range of 3200–3600 cm^−1^ was selected for investigation due to differences in the chemical structures between PS and PVA (the presence of a hydroxyl group). Thermal transition behavior of polymers were evaluated using modulated differential scanning calorimetry (M-DSC, DSC Q200; TA Instruments Inc., New Castle, DE, USA). A contact angle meter (DM 500; Kyowa Interface Science Co., Ltd., Saitama, Japan) was used to measure the surface interactions between molten polymers and AAO surfaces, and between water droplets and polymer surfaces. The friction coefficient of polymer surfaces at the macroscale was evaluated by a friction and wear tester (EFM-III-F; Orientec Co., Ltd., Tokyo, Japan) using stainless steel materials. The friction test conditions were evaluated using an applied force of between 21–40 N with the rotational speed of 10 rpm for 30 s. All of the tests were performed at ambient conditions (25 °C, relative humidity 35%). The surface mechanical properties at the nanoscale were evaluated by nanoindenter (G200; Agilent Technologies Inc., Santa Clara, CA, USA).

## 3. Results and Discussion

### 3.1. Characterization of AAOs 

Figure 2 shows SEM images of the surface morphologies of fabricated AAO templates. The AAO templates consist of dense arrays of cylindrical nanopores displaying abundant honeycomb–like porous cavities caused by the formation of highly ordered hexagonal closely packed morphology, formed by the two–step anodization process. The narrow pore size of the AAO templates can be fabricated under our experimental conditions. Figure 2a shows the prepared AAO^1^ template with a 50 nm diameter pore, 25 nm inter–pore distance, and 120 µm depth. For the AAO^2^, Figure 2b shows a template with 100 nm pore size, 50 nm inter–pore distance, and 130 µm depth. In this experiment, it is noted that increased pore size is mainly influenced by increased applied electrical potential. The slight increase in pore depth occurs because of the increased anodizing time and concentration of the electrolytic solutions. In this study, the AAO^2^ templates were selectively used to examine the phase separation behaviors and structural transformation behaviors of polymer nanostructures. Figure 2c shows a cross–sectional side view of AAO^2^ nanocavities of 100 nm pore diameter. The SEM micrograph depicted in Figure 2c confirms that an AAO^2^ having a smooth surface can be prepared in this study. The pore wall of the AAO^2^ shows an almost perfectly smooth surface and corresponds to the expected dimensions and invariance of the pore diameter through the length of AAO^2^ templates.

### 3.2. Fabrication of One–dimensional Nanostructures by Thermal Nanoimprint

In this part, the flowability and confinement effects of PS^1^ and PS^2^ at the nanoscale will be clarified. Figure 3 shows representative SEM images (side view) of PS^1^–VANs and PS^2^–VANs obtained by thermal nanoimprint using the AAO^1^ template. For the each of the SEM images, PS^1^–VANs and PS^2^–VANs were produced using the imprinting conditions of 180 °C, 5 MPa, for 30 min. The crucial point for performing the thermal nanoimprint is that the imprinting temperatures must be above *T_g_* or *T_m_* of the polymers which is sufficient for chain mobility and adequate for flowability at the nanoscale in AAO cavities. In Figure 3, PS^1^–VANs and PS^2^–VANs with lengths of 70 and 40 µm respectively can be produced. The obtained VANs, having bundle–like structure, exhibit orderly array structures but the tendency to agglomerate. The appearance of agglomerated nanostructures is explainable by surface adhesion induced between individual nanostructures. Generally, the quality of agglomerated nanostructures was enhanced by using lower aspect ratios and higher interpore distances in AAO templates.

As Figure 3, the difference in the length of VANs due to the different flowability of the PS precursors in the range of 160–220 °C is shown, with fixed imprinting pressure and imprinting time of 5.0 MPa and 30 min respectively. The longer length of PS^1^–VANs is compared with PS^2^–VANs, due to the higher MFR and lower *T_g_* of PS^1^. The lower value of *T*_g_ indicated higher chain mobility of polymer due to larger free–volume of the polymer chain packing. The PS^1^ showed the higher flowability within AAO^1^ templates as compared with PS^2^. The flowability of the polymers was enhanced by increased imprinting temperature due to the reduction of polymer viscosities, and decreased surface tensions between polymer melts and AAO templates [11,12,13,14,15]. However, the length of VANs did not increase at temperatures above 180 °C because the polymer was easily ejected out from the AAO nanocavities under the high imprinting pressure and relatively low viscosity. PS^1^–VANs and PS^2^–VANs of about 15–72 µm long were obtained depending upon the imprinting conditions.

The flowability at the nanoscale of the molten polymers in the AAO templates driven by the imprinting process is in agreement with the Hagen–Poiseuille expression [15,50]. The length (H) or distance the polymer flowed through AAO can be estimated as shown below. 

(1)$$H=2r\sqrt{\frac{\left(P+\frac{2\gamma \cos\theta }{r}\right)t}{32\eta }}$$

This equation shows that the factors related to flowability in confined spaces are imprinting pressure (P), radius of the nanochannel (r), surface tension of the molten polymer (γ), contact angle of the polymer on AAO pore wall (θ), viscosity of the molten polymer (η), and infiltration time (t). Our previous study revealed that the reduction of apparent viscosity estimated by Equation (1) may be the result of shear rate, wall slip, and the pressure used in the imprinting process [15].

To avoid the effects of confinement on the phase separation and transformation behaviors at the nanoscale, therefore, we investigated this change in molecular free–volume by analyzing the thermal properties of obtained PS–VANs as compared with PS bulk films. Specimens were obtained by imprinting conditions of 180 °C, 5 MPa, for 30 min with different AAO pore size and *T_g_* of polymers. Figure 4 shows the DSC heating thermograms of the PS bulk films and PS–VANs. The results indicate a change in molecular free–volume was found in the case of PS^2^–VANs when using AAO^1^ templates. In the case of PS^2^, an increase in *T_g_* of the PS^2^–VANs to 104 °C is found deviating from the PS^2^ bulk film precursors having *T_g_* of 100 °C. The increase in *T_g_* of PS^2^ after the imprinting process using the AAO^1^ template due to the nanoconfinement, influences the reduction of the free−volume or the increase in polymer chain packing density caused by the compression, and shortens each polymer chain distance. It also might be caused by the applied force from the imprinting pressure induced the reduction of free–volume polymer in the AAO nanopore. Moreover, the polymer chains at the interface have a strong interfacial interaction with the AAO pore walls that can induce the orientation change of polymer chains [51,52] and also immobilize the polymer chain at the interface, especially under rapid cooling conditions. The result shows that using AAO^1^ template induced the confinement effect on the reduction of the free–volume of PS^2^. Hence, the use of PS^1^–AAO^2^ imprinting was selected to investigate the phase separation and structural transformation behaviors at the nanoscale.

### 3.3. Phase Separation and Structural Transformation of Polymer Blends at the Nanoscale

Phase separation and the structural transformation behaviors of PS^1^/PVA blends confined in AAO^2^ nanochannels are experimentally investigated herein. The effects of domain size and the mass ratios of PS^1^/PVA blends were investigated to clarify characteristic behaviors confined in the AAO^2^ template, which influence the obtained morphologies of PS^1^ nanostructures. Figure 5a,A and b,B show the SEM images of the cross–sectional PS^1^/PVA blends prepared by the advanced high–shear process. PVA dispersed phases with the domain size in the range of nanodomain and microdomain were prepared with the difference in PVA mass ratios of 10 and 30 wt% respectively. In the case of PS^1^/PVA (90/10 wt%) blend, PVA nanodomain was formed having a diameter in the range of 10–30 nm. For the PS^1^/PVA (70/30 wt%) blend, PVA microdomain was produced showing the smallest and average diameters of 300 nm and of 30 µm, respectively. Figure 5c,C presents the mapping image of the ATR of the PS^1^/PVA film precursors. These results indicate that the uniform dispersion of the PVA in the PS^1^ matrix was able to be prepared by using the high-shear process. However, the non–uniform area indicated signal distortion due to scattering of Infrared radiation (IR).

Figure 6A illustrates the top view images of PS^1^–VANs surface morphologies produced by thermal nanoimprint process. The obtained smooth surface of the PS^1^–VANs can be prepared using the neat PS^1^ film precursors. PS^1^–VAPNs can be produced using the PS^1^/PVA (90/10 wt%) precursor films followed by the selective removal of PVA segment as shown in Figure 6B,b. In the case of the PVA microdomain, the PS^1^–VANs with the nanopeapods–like form was replicated consisting of nanorods and nanospheres (See Figure 6C,c). The microholes on the imprinted film surface were observed because of the removal of the PVA segment. Transformation of PS^1^ nanostructures from nanocolumns into the nanorods and the nanospheres was induced by the Rayleigh instability due to the interfacial interaction of PS^1^ and PVA. The reduction of dimensionality from 1D into 0D is the intention of decreasing the surface energy of the system. In this experiment, the influence of imprinting conditions (temperature and time) were investigated to clarify the phase separation and structural transformation behaviors of the polymer nanostructures confined in AAO^2^ templates.

Formation of the porous PS^1^ nanostructured surfaces was able to be generated by the surface–induced phase separation of the PS^1^/PVA (90/10 wt%) blend in AAO^2^ templates. Figure 7a–d illustrates the influence of the imprinting temperatures (120–180 °C) on the formation of pores on the nanostructured surfaces of PS^1^. The visible pores were able to be imaged at the imprinting temperature at 180 °C. At this temperature, the formation of nanopores was caused by; (1) the flowability of PVA at melting state (*T_m_* = 161 °C) and (2) the surface–induced phase separation due to the affinity of the hydrogen bond between AAO and PVA. Interfacial interaction between PVA-AAO^2^ also promotes the coarsening behavior owing to the hydrogen bond. Therefore, the PVA droplets in the PS^1^ matrix were formed on the pore wall of the AAO^2^ template. Moreover, the reduction of PS^1^ viscosity with increasing imprinting temperature can promote the surface-induced phase separation. Figure 7e illustrates the reduction of the polymer melt droplet angles of the PS^1^/PVA (90/10 wt%) blend with increasing temperature. At the temperature of 180 °C, it was found that the polymer melt droplets had a contact angle lower than 90^°^ which indicates the oxophilicity with AAO^2^ templates. Hence, the surface interaction between polymers and AAO becomes the important role in the phase separation at the imprinting temperature of 180 °C. Furthermore, thermal-induced flowability of PVA might enhance the formation of PVA droplets on the AAO^2^ pore wall due to the low viscosity at above melting temperature. 

Surface-induced phase separation at the macroscale has been investigated with the annealing process of PS^1^/PVA (90/10 wt%) blend films covered by the AAO template (upper) and aluminum plate (lower) at the temperature of 120–180 °C for 30 min. The surface chemical properties of the annealed films were simulated using the ATR mapping mode. By the result, the surface film of the polymer blend precursor has the composition of PVA of about ~10% (% hydroxyl group). After the annealing process at 180 °C, the ratio of PVA on the blend film surface adhered on the AAO side was reached about ~60–80% but showed the lower ratio on the aluminum side. The increase in the ratio of PVA on the annealed PS^1^/PVA blend film surface indicated the strong surface interaction between PVA and AAO above the melting temperature of PVA, as previously explained for the formation of nanoporous results. This can be confirmed that the formation of the porous PS nanostructured surface is caused by surface-induced phase separation at the imprinting temperature of 180 °C (See Figure 8).

According to the coarsening behavior, influences of coarsening temperature (with the function of viscosities and surface tensions) and coarsening times are the crucial parameters to the formation of the porous nanostructures. In this part, the influence of the imprinting time on the formation of the porous nanostructured surfaces will be investigated. The imprinting temperature and pressure of 180 °C and 5 MPa were performed in the experiment. Figure 9 presents the surface appearances of PS^1^ nanostructures obtained from the imprinting time of 15–120 min. With the long imprinting time of about 90–120 min, we found the breakup of the PS^1^ nanocolumns into nanofragments due to the agglomeration of the PVA droplets during and after the coarsening behaviors. The agglomeration of PVA droplets depends on the time of diffusion and mass transfer of the PVA.

Jiun Tai Chen et al [19] have explained the surface–induced phase separation behaviors by the coarsening process between PMMA and THF confined in AAO templates. The explanation reveals that the coarsening process progressed by the three possible mechanisms consisting of the Ostwald ripening, coalescence, and hydrodynamic flow. The Ostwald ripening was widely used to clarify the phase separation of the polymer solutions. The driving force of this mechanism is to minimize the interfacial energy between the polymer–rich phase and solvent-rich phase or the PS^1^–rich phase and PVA–rich phase for this case. The molecules of the tiny droplets of the dispersed phase dissolve due to the higher curvatures and precipitate on the surface of the large droplets. The reduction of the total interfacial area results in the decrease in the interfacial energy. The asymptotic power–low relation was proposed to explain the domain size of the droplets as in the following equation,
(2)$$d\approx {\left(D\xi \right)}^{1/3}t^{1/3},$$
where d is domain size of droplet, D is the diffusion coefficient, ξ is the correlation length, and t is the coarsening time. 

The second mechanism of the coarsening process is caused by the coalescence of the two droplets to form the larger droplet. The impinging of two droplets caused by the translational diffusion promotes the mass transfer, and then the larger single droplet is formed. The droplet domain size relates to time as in Equation (3)
(3)d≈(kBTη)13t13
where $k_{B}$ is the Boltzmann constant, T is the temperature, and η is the viscosity. In this study, the viscosity of molten polymers used was reduced with increasing temperature referred to the viscosity Arrhenius model [11].

The third coarsening mechanism caused by the hydrodynamic flow of the fluid mixtures. The gradients of pressure along the axis of the cylindrical pore play the role of the flowability from a narrow to a wide region that induces the bi–continuous of the two polymers formed during the phase separation. The growth of the droplet domain is related linearly with time as in the following equation. Figure 10 simulates the phase separation caused by the coarsening behavior.
(4)$$d\approx \frac{\sigma }{\eta }t$$
where σ is the surface tension.

In the case of the PVA microdomain, Figure 7C,c illustrates the morphologies of PS^1^ nanostructured surfaces using PS^1^/PVA (70/30 wt%) blend precursor. After the removal of the PVA segment, we found PS^1^ nanostructure arrays with microholes. The appearance of microholes is caused by the PVA microdomain size of the film precursor. At the microholes area, we found PS^1^ nanostructures with nanorods and nanospheres (See Figure 11). The transformation of the PS^1^ nanostructures is the result of the surface interaction between PS^1^ and PVA in the AAO^2^ templates. The PS^1^ encapsulated by the PVA–rich phase (30 wt%) adjoined with the AAO^2^ pore wall, and the interfacial interaction between PS^1^ and PVA lead to the structural transformation of the polymer nanostructures, which was driven by the Rayleigh instability within AAO^2^ templates. Rayleigh instability is the common transformation phenomena when the columns of water are falling and breakup into water droplets. The instability is the result of the undulation of the liquid cylinder surface at low surface tension. Herein, the influences of imprinting temperatures (120–180 °C) and imprinting time (30–120 min) have been investigated. It was found that the number of the spherical structures increased when increasing the imprinting temperature and imprinting time. With increasing the imprinting temperatures, Rayleigh instability caused by the decrease in viscosities that enhances the degree of structural transformation of the PS^1^ nanocolumns and converted to the nanorods and nanospheres. The increased imprinting temperature also resulted in the increasing rate of mass transfer and the kinetic phenomenon of the transformation. The increase in the immiscibility of the polymer blends due to the rise in the temperature that influences the increasing heat of mixing. Therefore, it was found that the number of nanorods and nanospheres increased when elevating the imprinting temperatures.

In terms of the influence of the imprinting time, we found the similarity in the structural transformation behaviors when increasing the imprinting temperatures. The number of nanorods and nanospheres increased with increasing the length of imprinting time as shown in Figure 11. However, the result reveals that the agglomeration of the PS^1^ segment has not much effect in this case. Jiun Tai Chen et al. explained clearly the transformation of the nanostructures driven by the Rayleigh instability with the characteristic time scale that agrees with their experimental results. The Rayleigh instability was able to explain this by Equations (5) and (6) as below [24,25]. Figure 12 simulates the phase separation caused by the coarsening behavior.

In the case of fluid viscosity is neglected, the break up time as the characteristic time scale driven by the Rayleigh instability is
(5)$$\tau ={\left(\rho R_{0}/\gamma \right)}^{1/2}$$
where ρ is density of the fluid and $R_{0}$ is the initial radius of fluid cylinder. 

In the case of viscoelastic materials, the characteristic time is given by

(6)$$\tau =\eta R_{0}/\gamma$$

### 3.4. Surface Properties 

Surface properties of polymers were evaluated using the water droplet angle measurement, macroscale friction test, and nanoindentation. The five types of polymer films used to investigate were; (1) neat PS^1^, (2) neat PVA, (3) PS^1^/PVA (90/10 wt%) blend, (4) PS^1^–VANs, and (5) PS^1^–VAPNs. The imprinted samples were obtained using the imprinting condition of 180 °C, 5 MPa for 30 min (nanostructures length 70 ± 10 µm with the remained film thickness 40 ± 10 µm). 

The results of the water droplet angle measurement was presented in Figure 13. The flat surfaces of PS^1^ and PVA films with the droplet angle of water about ~89.8° and ~40°–50° (initial droplet) were obtained. The water–soluble properties of the PVA influence the instability of the water droplet shape and its angle as shown on the shadow side of the water droplet. The water droplet angle of PS^1^/PVA film with a value of about ~65.6° can be measured. PS^1^–VANs surface shows the water droplet angle reach ~130.1°. The great increase in water droplet angle as compared to the flat films is the result of the physical roughness of the nanostructures and the reduction of the contact area between polymer and water droplet. Moreover, the PS^1^–VAPNs surface has a slight increase in the water droplet angle to ~140.1° as compared with the PS^1^–VANs due to roughness at the nanoscale.

Surface friction properties at macroscale demonstrated the same tendency with results of the water droplet angle measurement. The measured friction coefficients was placed in Figure 14. The PS^1^–VAPNs surfaces exhibit the higher friction coefficient (~0.53–0.55) as compared with PS^1^–VANs surface (~0.37–0.42). The results indicate the roughness at the nanoscale influences the changes in surface friction at the macroscale, and the surface energy in the water droplet angle measurement as in the explanation above [53,54,55,56]. In addition, the friction coefficient of PS^1^–VAPNs can be enhanced when it was switched to the wetting condition. This result is due to the water bridge–mediated contact formation induced by the solid–solid contact between the contact elements and the porous surface of PS^1^–VAPNs [9,10].

By the results of the nanoindentation test, the PS^1^–VAPNs show a lower surface hardness and surface modulus as compared with PS^1^–VANs due to the scaling down or the size reduction effect of the material structures after the removal of the PVA segment [57]. Hence, the size reduction has the result of a decrease in the energy absorption capacity and the destruction of the nanostructures becomes easier. The surface hardness of 0.6 and 0.8 MPa and surface modulus of 0.03 and 0.08 GPa of the PS^1^–VAPNs and PS^1^–VANs can be respectively obtained using the indentation test. Figure 15 presents the nanoindentation test results of the imprinted nanostructures.

## 4. Conclusions

In summary, phase separation and the structural transformation behaviors of the PS^1^/PVA blend in AAO template (100 nm diameter) for fabricating the PS^1^–VAPNs has been investigated. The replication process used is the thermal nanoimprint lithography. In this study, the different behaviors of the phase separation and the structural transformation were the result of the domain size and the mass ratios of PVA precursors. 

In the case of the PVA nanodomain (PS^1^/PVA 90/10 wt%), formation of the PVA droplets on the AAO pore wall due to the surface–induced phase separation behavior, resulting in the porous surface which occurred after the removal of PVA. The formation of pores on the PS^1^ nanostructured surface occurred at imprinting temperatures above the melting temperature of PVA. This result indicated the high flowability of the molten PVA has an important role in the mass transfer of the phase separation. The increase in imprinting time promotes the agglomeration of PVA droplets conductive to the breakup of PS^1^ columns into nanofragments. 

With the PVA microdomain precursor (PS^1^/PVA 70/30 wt%), the results demonstrated the transformation behaviors of PS^1^ nanostructures; PS^1^ nanocolumns transform into the nanopeapods consisting of nanorods and nanospheres. The structural transformation from 1D to 0D is induced by Rayleigh instability due to the surface interaction between PS^1^ and PVA. The influences of the increases in the imprinting time and the imprinting temperature resulted in an increased number of nanorods and nanospheres due to enhancing the rate and duration of the mass transfer, and the decrease of polymer viscosities.

For the surface properties, the formation of pores on nanostructured surfaces of PS^1^ has the result of increasing both the water droplet angle and the friction coefficient due to the increase in surface roughness at the nanoscale. Moreover, the friction coefficient of VAPNs can be increased when switchable to the wetting condition. However, we found the reduction in the surface hardness and surface modulus caused by the scaling down effect.

## Figures and Tables

**Figure 1 polymers-11-01039-f001:**
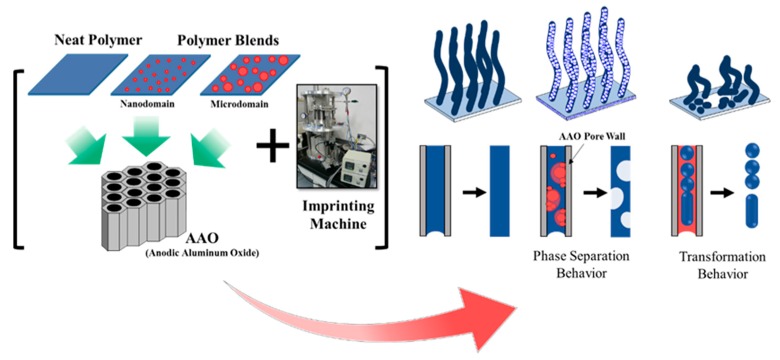
Fabrication of one-dimensional nanostructures by thermal nanoimprint.

**Figure 2 polymers-11-01039-f002:**
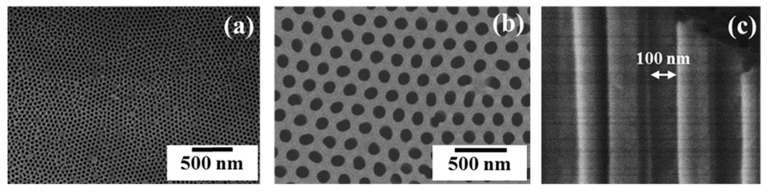
Scanning electron microscopy (SEM) images of surface morphologies of (**a**) AAO^1^ (**b**) AAO^2^ and (**c**) cross−sectional AAO^2^.

**Figure 3 polymers-11-01039-f003:**
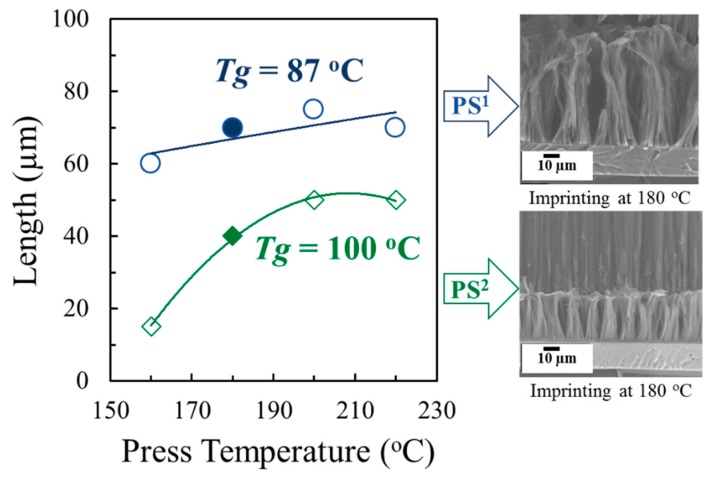
Effect of imprinting temperature on length of PS–VANs (SEM images refer to the solid symbols).

**Figure 4 polymers-11-01039-f004:**
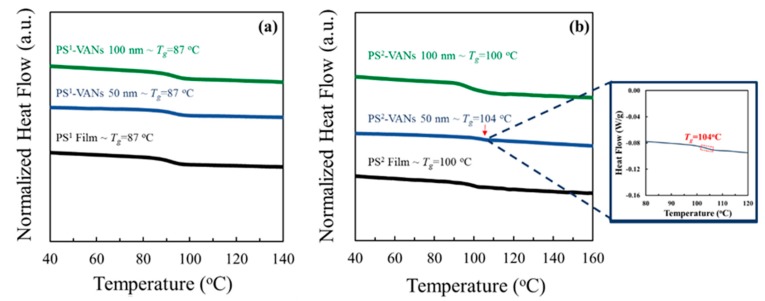
Differential scanning calorimetry (DSC) thermograms of polymer films and nanostructures of (**a**) PS^1^ and (**b**) PS^2^.

**Figure 5 polymers-11-01039-f005:**
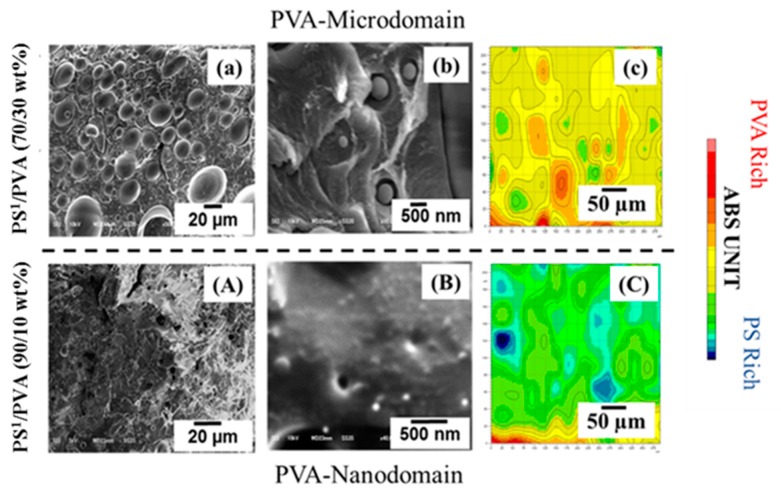
Surface morphologies and surface chemical analysis of PS^1^/PVA blends; (**a**,**b**) SEM images of PS^1^/PVA (70/30 wt%), (**A**,**B**) SEM images of PS^1^/PVA (90/10 wt%), (**c**) ATR mapping of PS^1^/PVA (70/30 wt%) and (**C**) ATR mapping of PS^1^/PVA (90/10 wt%).

**Figure 6 polymers-11-01039-f006:**
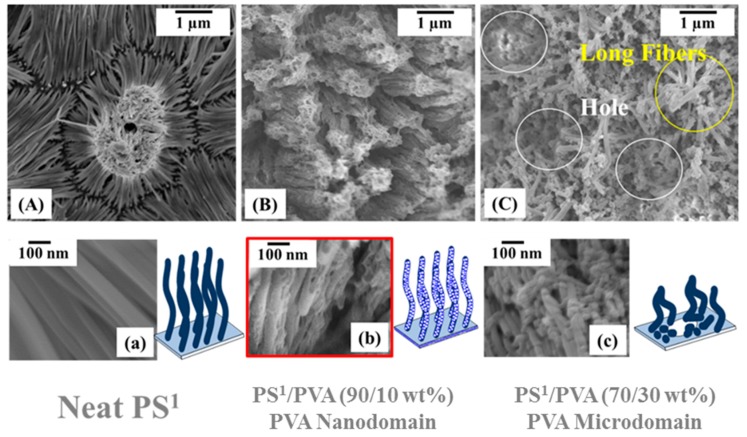
Surface morphologies of (**A**,**a**) PS^1^—VANs (**B**,**b**) PS^1^—VAPNs and (**C**,**c**) PS^1^ nanopeapods.

**Figure 7 polymers-11-01039-f007:**
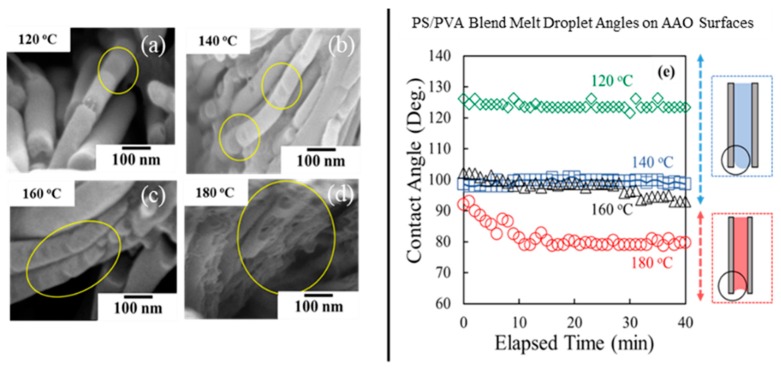
Effect of imprinting temperature on formation of nanoporous (left); (**a**) 120℃, (**b**) 140 ℃, (**c**) 160 ℃, (**d**) 140 ℃, and (**e**) PS^1^/PVA melt droplets on AAO surfaces (right).

**Figure 8 polymers-11-01039-f008:**
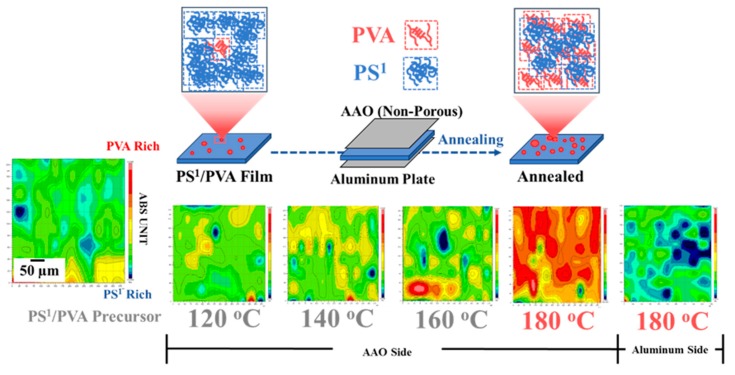
Annealing behaviors of PS^1^/PVA (90/10 wt%) blends.

**Figure 9 polymers-11-01039-f009:**
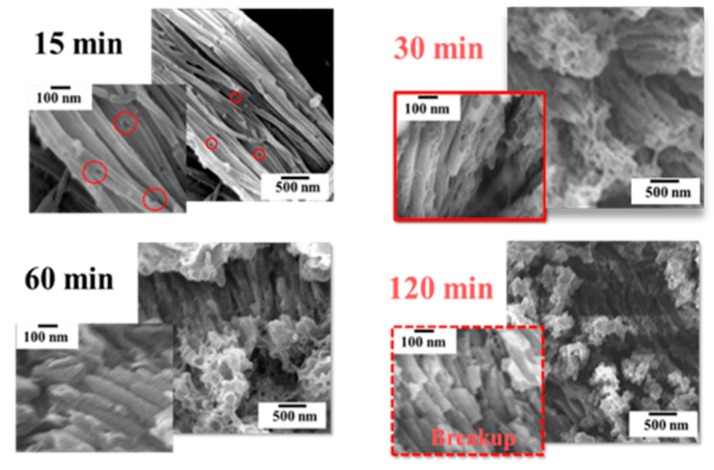
Effect of imprinting time on the obtained PS^1^ nanostructures.

**Figure 10 polymers-11-01039-f010:**
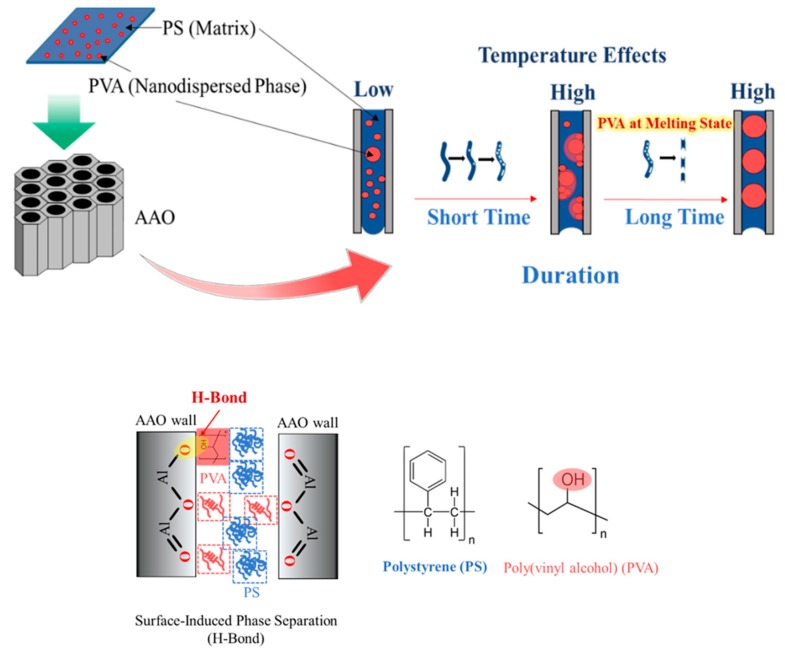
Formation of polymer droplets caused by surface−induced phase separation and coarsening behaviors.

**Figure 11 polymers-11-01039-f011:**
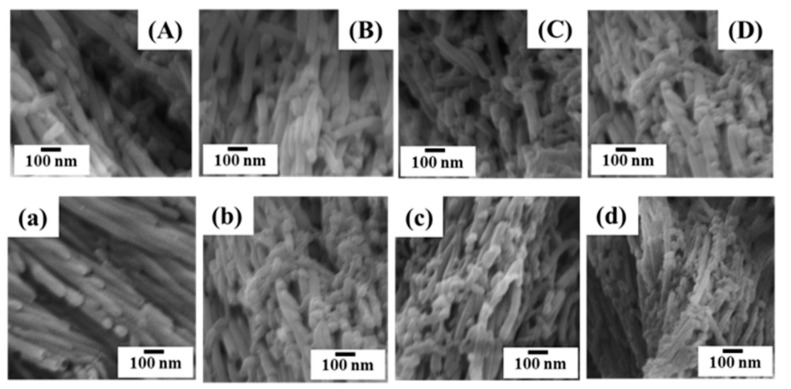
SEM images of PS^1^ nanostructured arrays obtained by PS^1^/PVA (70/30 wt%); (upper; imprinting time 30 min) with the imprinting temperatures of (**A**) 120 (**B**) 140 (**C**) 160 and (**D**) 180 °C, lower; imprinting temperature 180 °C with imprinting times of (**a**) 15 (**b**) 30 (**c**) 60 and (**d**) 120 min).

**Figure 12 polymers-11-01039-f012:**
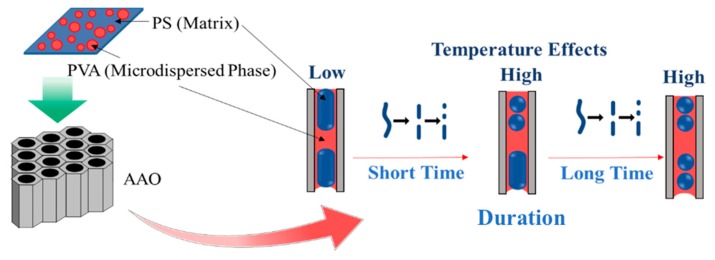
Transformation of polymer nanostructures driven by the Rayleigh instability.

**Figure 13 polymers-11-01039-f013:**
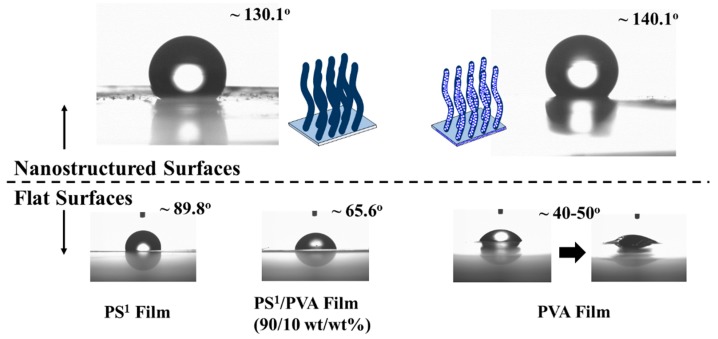
Water droplets on the polymer surfaces.

**Figure 14 polymers-11-01039-f014:**
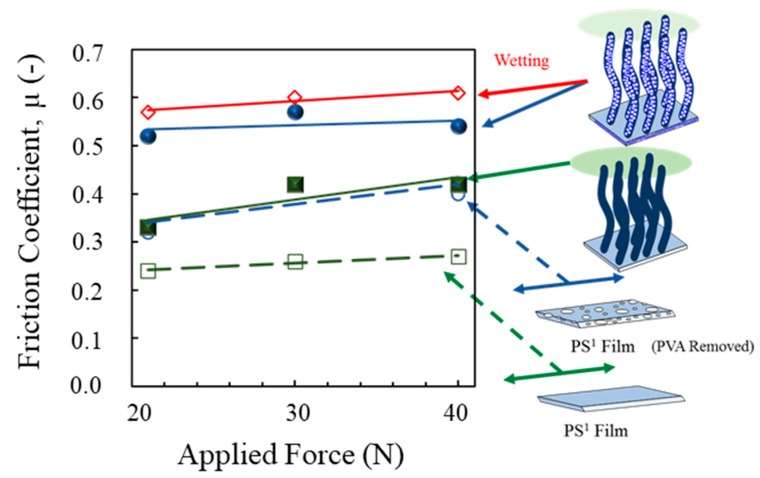
Friction coefficients of polymer surfaces

**Figure 15 polymers-11-01039-f015:**
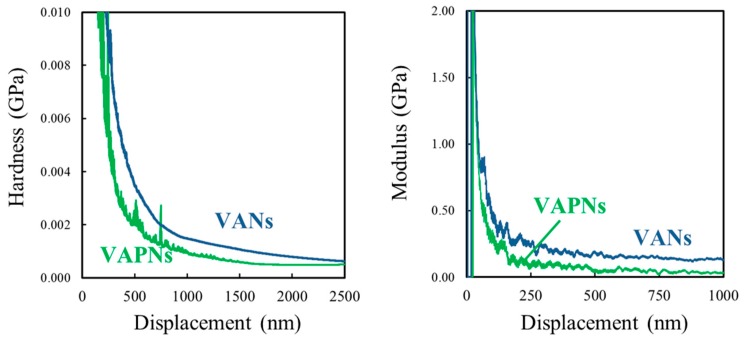
Nanoindentation test results.

**Table 1 polymers-11-01039-t001:** Anodizing conditions. Anodic aluminum oxide (AAO).

Anodization	AAO^1^		AAO^2^	
	1st	2nd	1st	2nd
Electrical Current (A)	12.1	12.1	12.1	12.1
Applied Voltage (V)	40	40	60	60
Oxalic Acid (M)	0.2	0.2	0.3	0.3
Time (h)	17	24	17	48
Temperature (°C)	10	10	5	5

**Table 2 polymers-11-01039-t002:** Imprinting conditions.

Materials	Temperature (°C)	Pressure (MPa)	Press Time (min)	Obtained Nanostructures
PS^1^	160−220	5	30	Nanopillars
PS^2^	160−220	5	30	Nanopillars
PS^1^/PVA (90/10 wt%)	120−160	5	30	Nanopillars
PS^1^/PVA (90/10 wt%)	180	5	15–30 60–120	Porous Nanopillars Nanofragments
PS^1^/PVA (70/30 wt%)	120−180	5	15−120	Nanopeapods

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
