# Peer review of "Replication of Mesoscale Pore One-dimensional Nanostructures: Surface-induced Phase Separation of Polystyrene/Poly(vinyl alcohol) (PS/PVA) Blends"

_polymers, 2019, doi:10.3390/polym11061039_

Reviewer 1 Report

The paper is interesting and covers a novel approach to obtain nanostructures with many functional properties. The paper is written in a plain way but some minor improvement in the English style are needed. As an example the phrase "Furthermore, the thermally induced of AAO2 pore wall might  promote the formation of PVA droplets due to the higher thermal sensitivity agree with higher chain 279 mobility of PVA at the melting state." is not clear and should be rewritten. 

In addition the overall organization of the paper is not well arranged with long discussions and no clear graph summarizing the overall results. A table summarizing the effect of the different processing parameters reported in table 2 vs the final structures obtained should be added. 

Paragraph 3.1 should be renamed into Characterization of AOOs. In addition, authors should explain here why only AOO2 was used in the paper. 

Author Response

May 9, 2019

Dear Reviewer of Polymers:

Thank you for the opportunity to revise our manuscript of “Replication of Mesoscale Pore One-dimensional Nanostructures: Surface-induced Phase Separation of PS/PVA Blends” (polymers-462850). We appreciate the careful review and constructive suggestions. It is our belief that the manuscript is substantially improved after making the suggested edits.

Following this letter are the editor and reviewer comments with our responses in italics, including the respond to comments and the slight improvement of writing. The revision has been developed in consultation with all co-authors, and each author has given approval to the final form of this revision.

Thank you for your consideration. 

Sincerely, 

Paritat Muanchan,

Author

Reviewer 2 Report

This is a nice contribution. I favor acceptance. English editing is needed.

Author Response

(The authors gave the same response as above.)

Reviewer 3 Report

Ito and coworkers present some interesting results for a means to produce 1D Nanostructures of PS based materials.  The approach (and much of the discussion) leans quite heavily on the earlier work of Chen (ref 19) but has some novelty in the use of a second polymer to control the structures and does show some very elegant structures, which may have some interesting properties.  I find it difficult to judge how significant a step it is to achieve similar structure by nanoimprinting, but would say that the originality/novelty depends on this aspect.

Overall, I think that the paper could be of interest to readers of Polymers, but I am concerned by the limited characterisation of the materials, and rather limited evaluation of the results as this undermines the scientific quality.  

The introduction covers the relevant background material well enough and is generally well referenced.  (One exception to this is the claim (p2 line 45) re. energy storages (sic) supramolecular membranes etc could do with references).

My main concern is with the materials characterisation.  The polymers are barely defined at all.  There are Tg values for the polystyrenes (in one case so low that I wonder if it is oligomer or polymer) and melt-flow rate characterisation, using different temperatures for each sample.  This precludes any meaningful comparison of the samples.  Since the nano imprinting and phase separation phenomena, which are critical to the results depend on viscosity and molecular weight respectively temperature dependent viscosity data and Mw, Mn should be given as a minimum.  The characterisation of the PVA is similarly lacking - degree of hydrolysis will impact crystallinity and Tg, Tg is also very sensitive to water content/humidity, but these are also omitted.

The experimental description is too brief to enable anyone to reproduce these results, which is a shame because they are interesting.  What were the cooling conditions for palletisation (line 123)?  This will clearly impact on the initial coarsening.  Exactly what samples were the DSC carried out on?  Was the AAO support still present?  What was the material in contact with the test sample surface in the friction measurements (line 155) - stainless steel or something else?  I think that the solid points in figure 3 are solid because these may correspond to the images to the right of the figure, but this detail is not apparent from the legend.

Equation 1 - would be meaningful if viscosity data were included, but without this and at least an estimate of surface energy, it adds nothing.

I find the discussion on molecular orientation based on a single outlying Tg value unconvincing.  For significant chain stretch/reorientation, having impact on free volume and therefore Tg, the flow rate into the pore would have to be fast compared to the terminal or stretch relaxation times, and the sample would have to be quenched rapidly compared to this time too.  I am not convinced that this is possible or will have happened, given that two instruments, one for hot pressing and one for cold pressing were used.  Was the Tg measured with the sample still in the template AAO?  (This could increase the Tg of the smaller pore sample via anomalous surface Tg effects c.f. 10.1039/FD9949800219 Faraday Discuss., 1994,98, 219-230 and many others)  For the pressure effect to be credible as a source of the Tg increase there needs to be comparison to literature values of Tg vs sample pressure.

line 246 - These results indicated uniform dispersion of the PVA in PS...  - the figures clearly show just the opposite; that there are regions of high and low concentration evidenced by the color maps.

The discussion of PVA segregation around line 290 onwards is interesting, but would be much more informative if some measure or even estimate of the depth range of the ATR was given.  Presumably the composition change is for the depth-averaged composition over this measurement range.  Knowing this in relation to the pore depth can tell us if the PVA forms something like a pure layer over a short range, or a more diffuse form of enrichment over a greater range.

Figure 10 - I don't see why the PVA that initially wets the AAO interface is then displaced from the interface in the final image on the right hand side.

Equations 2 and 3 are introduced but not used and not useful without viscosity data.  Equations 5 and 6 are given, but not used.  These could potentially be used to answer the question: "Is viscosity significant in determining the Rayliegh instability", but instead they are redundant.  This contributes to the overall sense that the depth of analysis is lacking, which is a shame because the phenomenology is interesting. 

Line 401 - an interesting result for the hydrophobicity - perhaps worth mentioning that this is strong evidence for Cassie-Baxter wetting - would benefit from lit reference to contact angle of water on smooth PS surface.  (about 90 degrees, I think)

Figure captions need a bit more detail.  They should specify the measurement type e.g. SEM, FTIR etc. where appropriate

Overall, I think that there are some interesting and publishable results here, but that the work is  let down by the limited materials characterisation and critical evaluation of the results in the context of available literature.  

Author Response

May 9, 2019

Dear Reviewer of Polymers:

Thank you for the opportunity to revise our manuscript of “Replication of Mesoscale Pore One-dimensional Nanostructures: Surface-induced Phase Separation of PS/PVA Blends” (polymers-462850). We appreciate the careful review and constructive suggestions. It is our belief that the manuscript is substantially improved after making the suggested edits.

Following this letter are the editor and reviewer comments with our responses in italics, including the respond to comments and the slight improvement of writing. The revision has been developed in consultation with all co-authors, and each author has given approval to the final form of this revision.

Thank you for your consideration. 

Sincerely, 

Paritat Muanchan,

Author

Round  2

Reviewer 3 Report

The authors have addressed most of my concerns in their revised manuscript; therefore I would recommend publications subject to a few minor points of clarification:

Caption for figure 3 should state that images correspond to the solid points.

line 216 Reduction of apparent viscosity increases the flowability of the polymers

would connect better to the introduction if it made it clear how viscosity is related to melt flow index, since this is the measure of viscosity that the authors provide in lieu of viscosity measurements.

I still have concerns with the statement: "Figure 5 (C, c) presents the 247 mapping image of the ATR of the PS1/PVA film precursors. These results indicate that the uniform 248 dispersion of the PVA in the PS1 matrix was able to prepare by using the high-shear process." 

The images, (especially 5C) clearly show significant variation in color, which according to the scale bar correspond to variation in composition.  5C has features from dark blue PS-rich to dark red PVA rich, so it appears to be far from uniform on the measurement scale.

The authors respond, "

: This color in the mapping results is clearly uniform color for polymer blends. A bit not perfectly color distribution because of the light scattering.  "

The text makes no mention of light scattering, or indeed that the entire range of composition is spanned for a nominally homogeneous sample.  If light scattering is necessary to justify the interpretation, then it must be included (and the effect referenced) for the benefit of other readers; otherwise many will not believe the interpretation of the data that is presented.

The authors make a good point with their response:"

: As I mentioned before, viscosity of polymers usually followed Arrhenius equation due to the change in temperature. Viscosity was reduced with increasing temperature. We just need the trend of viscosity related to the formation of polymer nanodroplets, resulting in the formation of porous structures after removal of PVA segment. Also with the effects on the Rayliegh instability as well.

"

Perhaps it would be worth pointing this out in the paper to make this clear to the readers that viscosity is being controlled via temperature as it follows an Arrhenius (or WLF?) dependence.

Author Response

Dear Reviewer of Polymers:

Thank you for the opportunity to revise our manuscript of “Replication of Mesoscale Pore One-dimensional Nanostructures: Surface-induced Phase Separation of PS/PVA Blends” (polymers-462850). We appreciate the careful review and constructive suggestions. It is our belief that the manuscript is substantially improved after making the suggested edits.

Following attachment letter are the editor and reviewer comments with our responses in italics, including the respond to comments and the slight improvement of writing. The revision has been developed in consultation with all co-authors, and each author has given approval to the final form of this revision.

Thank you for your consideration. 

Sincerely, 

Paritat Muanchan,

Author
